# BROWSERAGENT: BUILDING WEB AGENTS WITH HUMAN-INSPIRED WEB BROWSING ACTIONS

## ABSTRACT

Efficiently solving real-world problems with LLMs increasingly hinges on their ability to interact with dynamic web environments and autonomously acquire external information. While recent research like Search-R1 and WebDancer demonstrates strong performance in solving web tasks, they heavily rely on additional tools to convert the interactive web environment into static text content. This is in contrast to human browsing behaviors, which involve diverse interactions with the browser, such as scrolling, clicking, and typing. In this paper, we propose BrowserAgent, a more interactive agent that solves complex tasks through human-inspired browser actions. BrowserAgent operates directly on raw web pages via Playwright through a set of predefined browser actions. We adopt a two-stage training (Supervised Fine-Tuning (SFT) and Rejection Fine-Tuning (RFT)) to improve the model's generalization abilities. Despite using significantly less training data than Search-R1, BrowserAgent achieves more competitive results across different Open-QA tasks. Additionally, we introduce an explicit memory mechanism to store key conclusions across steps, further enhancing the model's reasoning capabilities for long-horizon tasks. Notably, BrowserAgent-7B can achieve around 20% improvement over Search-R1 on multi-hop QA tasks like HotpotQA, 2Wiki, and Bamboogle. These results indicate that BrowserAgent can serve as a more advanced framework for more interactive and scalable web agents.

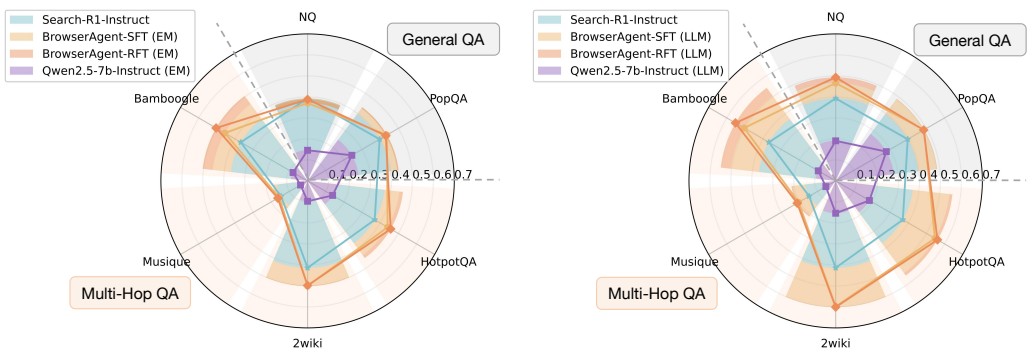

Figure 1: Evaluation Result of the Browseragent.

## 1 INTRODUCTION

Web agents (Lieberman et al., 1995; Yao et al., 2022) are autonomous systems that can interact with vast amounts of web information to make decisions and take actions to accomplish complex real-world tasks. It has been a long-standing research field to study how to automate the complex information-seeking process. Recently, Deep Research Agents (OpenAI, 2025) from ChatGPT, Grok, and Gemini demonstrated that current LLMs exhibit unprecedented abilities to solve highly complex real-world tasks (Hendrycks et al., 2025; Mialon et al., 2023). Following these commercialized deep research systems, there has been a surging interest in the open-source community to replicate their promising results. For example, Chen et al. (2025a); Li et al. (2025c); Song et al.

(2025); Jin et al. (2025a); Sun et al. (2025); Zheng et al. (2025); Liu et al. (2025a) have attempted various forms of SFT and RL approaches to post-train LLMs into web agents. However, these frameworks rely heavily on external tools like HTML-parser (e.g., Jina service) and summarizers (e.g., prompting GPT-4o to summarize the web page) to turn dynamic web information into static short-length documents. This will **(1) restrict the agents' abilities to interact with web pages to acquire in-depth information, and (2) incur a very high cost due to calling the additional tools.**

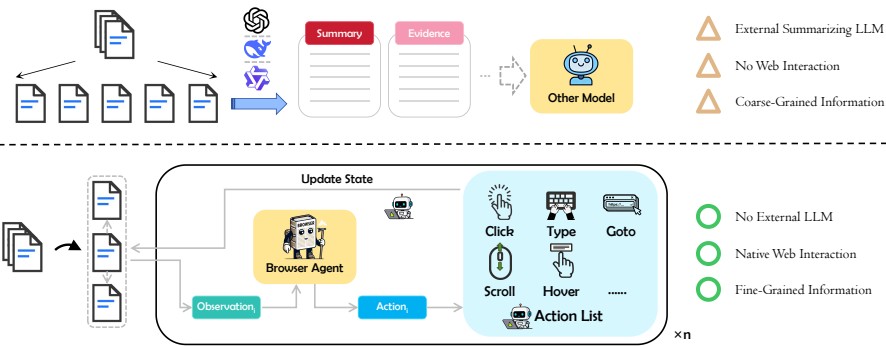

Figure 2: Comparison Between BrowserAgent and Traditional Deep Research Pipeline.

In contrast, humans exhibit highly interactive browsing behaviors when solving complex web tasks. For example, Humans perform actions like hyperlink clicking, form typing, and scrolling up/down through the browser to navigate through the web for information gathering. This allows humans to acquire more in-depth and native knowledge from the web without relying on additional summarization services. Inspired by this, we resort to browser automation frameworks like Playwright to enable these interactions with dynamic DOM elements, allowing LLM agents to share the same representation space and action set as humans do.

While browser-based agents enable rich interaction, prior works such as AssistantBench (Yoran et al., 2024) and WebArena (Zhou et al., 2024) primarily used them as evaluation-only benchmarks, because Playwright cannot be efficiently parallelized. This results in extremely low throughput ( 1–2 episodes/minute), making large-scale training infeasible. To overcome this, we develop a Ray-parallelized orchestration layer that scales to dozens of Playwright instances. On a single 32-core server, our system achieves 50+ episodes/minute, reducing the cost of browser-native data collection by over an order of magnitude (see Section 2.3 for more details). Building on this infrastructure, we explore strategies for constructing high-quality, behaviorally rich training datasets directly from live browser sessions. In contrast to methods that passively snapshot every intermediate state, our system selectively maintains a structured historical memory of key conclusions while prioritizing the current observation context—balancing long-term reasoning with real-time perception. This improves both sample efficiency and downstream agent performance. A vivid comparison is depicted in Figure 2.

We build **BrowserAgent**—a scalable, end-to-end browser-native agent framework that learns directly from real-time web interactions rather than relying on static content abstraction. BrowserAgent offers several advantages over prior work: It defines a minimal yet expressive set of atomic browser operations—such as scrolling, clicking, typing, and tab management—that closely align with real human behavior on the web. Rather than relying on LLM-based semantic abstractions, BrowserAgent interacts directly with raw webpage states, enabling fine-grained and compositional decision-making. BrowserAgent also departs from complex reinforcement learning strategies, instead using a lightweight two-stage training pipeline of SFT followed by rejection sampling. This simple yet effective approach achieves strong performance—especially in multi-hop reasoning tasks—while requiring less data and infrastructure. Furthermore, its iterative ReAct-style reasoning framework, enhanced with explicit memory, empowers the agent to perform multi-turn interactions and complex inference across information sources. With only 5.3K training samples, BrowserAgent can already demonstrate strong generalization and robustness. Our model can outperform Search-R1 (Jin et al., 2025b) significantly on a wide range of Open-QA tasks.

Through these experimental results, we demonstrate that BrowserAgent can serve as a good alternative approach to existing frameworks. Our work sheds light on how to build more scalable and interactive web agents for solving complex web tasks.

## 2 BROWSERAGENT

In this section, we present the detailed design of the BrowserAgent training methodology. This includes: (1) **defining a minimal yet expressive set of atomic browser operations to generate high-quality training data through interaction with dynamic web environments, ensuring that the agent learns how to engage with real-world web content;** and (2) **training BrowserAgent using efficient SFT and RFT to enable it to handle complex reasoning tasks.** Our approach focuses on leveraging real-time web interaction instead of relying on static datasets, providing a more flexible and scalable training framework. By systematically generating data from interactive search behaviors, we have successfully enhanced the model's problem-solving capabilities and its ability to handle diverse, real-world information.

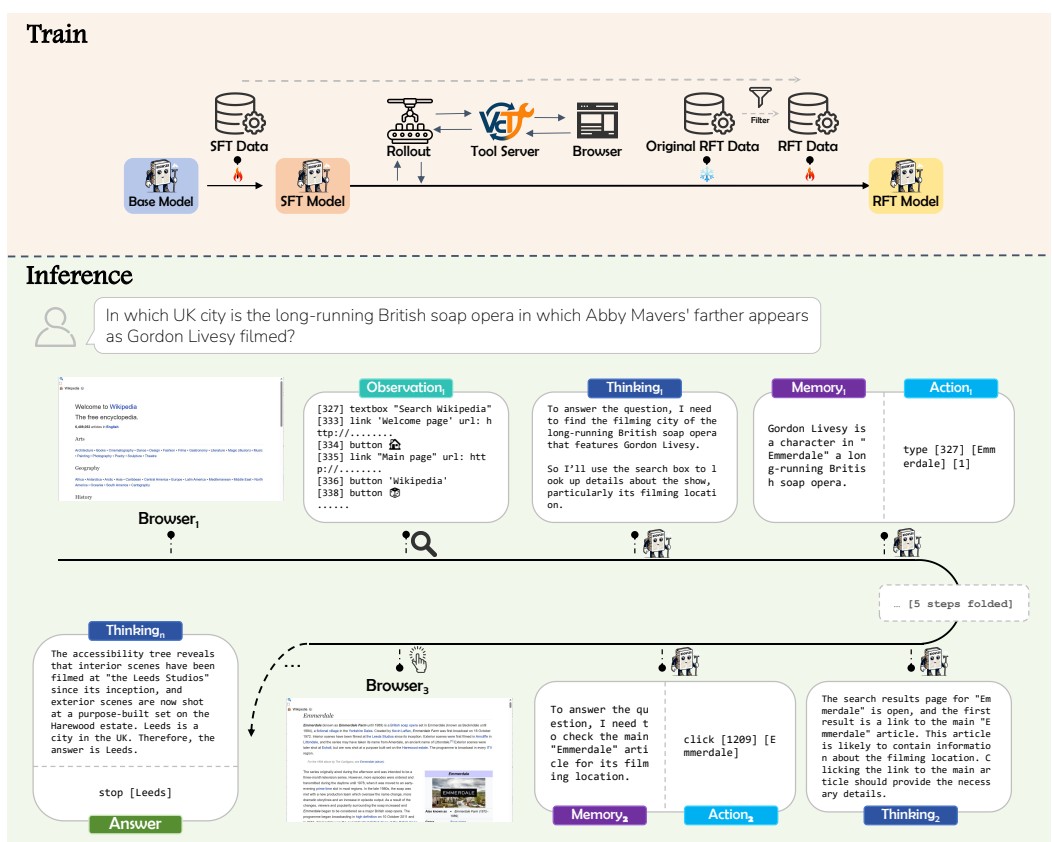

Figure 3: Overview of the Browseragent framework.

### 2.1 FRAMEWORK DESIGN

First, we simulate real human operations by defining four categories of actions: **page operation actions**, **tab management actions**, **URL navigation actions**, and **completion actions**. The BrowserAgent action categories and their corresponding command descriptions are provided in Table 1. These categories are designed to comprehensively cover the full spectrum of human–web interaction behaviors. Unlike other approaches, we do not rely on any additional models for web parsing and summarization (Wu et al., 2025). Instead, we simulate human browsing behavior through

Table 1: BrowserAgent Action Categories and Command Descriptions

| Human Operation Category | Action Commands | Descriptions |
|---|---|---|
| Page Operation Actions | `click(id, content)`
`hover(id, content)`
`press(key_comb)`

`scroll(down\|up)`
`type(id, content, press_enter_after=0\|1)` | Executes a click on the element specified by the id.
Hover the cursor over the element associated with the id.
Simulates the pressing of a key combination on the keyboard (e.g., Ctrl+v)
Scroll the page up or down.
Inputs content into the designated field (optionally followed by an Enter keystroke). |
| Tab Management Actions | `new_tab`
`tab_focus(tab_index)`
`close_tab` | Open a new, empty browser tab.
Switch the browser's focus to a tab using its index.
Close the currently active tab. |
| URL Navigation Actions | `goto(url)`
`go_back`
`go_forward` | Navigate to a specific URL.
Navigate to the previously viewed page.
Navigate to the next page (if a previous 'go_back' action was performed). |
| Completion Action | `stop(answer)` | Concludes the interaction by providing an explicit answer or designating the outcome as "N/A." |

direct interactions with the web page—such as scroll up/down actions in the page operation category—enabling the model to gradually develop the ability to autonomously locate required information and understand web content during training. This design brings multiple advantages. Fine-grained page operation and tab management actions not only enhance the model's understanding of web structures and its adaptability to multi-task scenarios, but also significantly improve the agent's generalization ability in complex web environments. Furthermore, by simulating human exploratory behaviors through actions like scroll up/down, the model is compelled to actively filter, locate, and integrate key information during real interactions. This avoids shortcuts such as relying on an additional large model for summarization, and gradually builds the model's capabilities in information retrieval, reading comprehension, and content integration throughout the training process.

In addition, **we simulate human search behavior through an iterative cycle of thinking–summarizing–acting.** Conceptually, this paradigm can be regarded as a variant of the ReAct framework (Yao et al., 2023b) augmented with a memory module, enabling the agent to maintain contextual continuity across multiple interaction turns. Specifically, both data generation and model inference are conducted through our automated multi-turn interaction process. We provide the system prompt and question, interact with the server to obtain observations, and then provide the prompt, question, and observations to GPT-4.1 (OpenAI, 2024), which outputs the (reasoning, action) pairs like ReAct (Yao et al., 2023b). Additionally, we require the model to record necessary conclusions. This process is iterated until the model outputs an answer to the question or the number of steps exceeds the limit. We keep track of historical actions and conclusions as memory in the input to help the model understand the completed actions and gather necessary information. The complete workflow is outlined in Algorithm 1. The prompt is provided in the Appendix B.3.

## 2.2 DATASET SYNTHESIS

To equip the model with comprehensive problem-solving capabilities, we selected the general question answering dataset NQ (Kwiatkowski et al., 2019) and the multi-hop question answering dataset HotpotQA (Yang et al., 2018). These two datasets cover basic factual questions and tasks that require complex reasoning and multi-step inference, helping the model learn richer reasoning paths and answering strategies in different types of problem scenarios. Our dataset consists of a total of 5.3K high-quality trajectories. Among them, 4K routes come from NQ and HotpotQA for simple question-answering tasks. The rest 1.3K trajectories are selected from HotpotQA to specifically improve the model's ability to tackle complex multi-hop reasoning tasks. All selected samples are processed through our automated multi-turn interaction collection process. The detailed procedure is described in Section 2.1.

For basic capability data, the system drives the large model to interact with the web environment for up to 6 steps, and for challenging data, it drives the large model to interact with the web environment for up to 30 steps. Every step is fully recorded, including the input prompt, model output, current page observation, and intermediate conclusions. This process ultimately generates detailed reason-

ing and decision-making trajectory data. Our case is provided in the Appendix B.4. Through this dual selection and collection strategy, our dataset not only ensures broad coverage of the model's basic capabilities but also specifically enhances its performance in complex, multi-hop reasoning scenarios, laying a solid foundation for subsequent agent training and evaluation.

### 2.3 ENVIRONMENT SETTING

Based on Verl-Tool (Jiang et al., 2025) and WebArena (Zhou et al., 2024), we implement a playwright-based tool to mimic the interaction with the browser. The observation is pure text-based, as an accessibility tree parsed by Playwright (Microsoft, 2020) (see example in Figure 3). To improve page readability, we do some rule-based post-processing such as merging consecutive text elements on the accessibility tree. To deploy Wikipedia (Karpukhin et al., 2020) locally, we adopt the offline Kiwix version with content updated to 2022. To standardize the interface and accelerate interactions, we employed the Verl-Tool framework with a Ray-based (Moritz et al., 2018) tool server as the top-level interface. All requests were routed through FastAPI (Ramírez, 2018), enabling uniform scheduling by the backend. Based on the design of Verl-Tool's tool server, each episode is assigned a unique trace_id to maintain the correspondence with its browser session, and the session is automatically terminated once a stop action is received. In our experiments, we deployed 64 concurrent Playwright browsers on a single machine with 32-CPUs. While the top-level scheduling is orchestrated by Ray (Moritz et al., 2018), the framework exhibits strong potential for scaling to much larger CPU clusters.

### 2.4 TRAINING

To enhance the model's ability to generate high-quality, structured answers, we propose a two-stage training framework comprising SFT and RFT, as illustrated in the Figure 3.

**Stage 1: SFT.** In the first stage, we perform SFT on the base model Qwen-7B-Instruct using 5.3K question-answer pairs generated via the method described in Section 2.2. This stage is designed to teach the model the desired answer format and to equip it with basic reasoning capabilities. The model is trained until convergence, and the checkpoint with the best evaluation loss is selected as the final SFT model. The evaluation loss curve is shown in the Appendix B.1. The primary goal of the SFT stage is to establish an initial understanding of the answer structure.

**Stage 2: RFT with Reasoning Path Selection.** In the second stage, we leverage the SFT model obtained from the first stage to construct training data for reinforced fine-tuning. Specifically, for each question in the training sets of NQ and HotpotQA, we sample four answers from the SFT model. We then use the EM metric for filtering, following the approach in (Xiong et al., 2025), to select instances where the model-generated answers include both correct and incorrect responses among the four candidates. From these instances, we choose the correct answer that contains the largest number of reasoning steps (Zhang et al., 2025), as more reasoning steps indicate deeper model thinking. This encourages the model to learn richer and more in-depth reasoning patterns. The selected samples constitute our Original RFT Data. To preserve the distributional characteristics of the original data and mitigate the issue of catastrophic forgetting of the answer format learned during the SFT stage, we randomly sample a portion of the SFT data and combine it with the filtered Original RFT Data to construct the final dataset used in the second-stage training. We then further fine-tune the SFT model on this mixed dataset, thereby enhancing its reasoning capability while maintaining its ability to generate well-structured answers. The specific data ratio and filtering strategy are detailed in the Appendix B.5.

## 3 EXPERIMENTS

### 3.1 BENCHMARKS

To comprehensively evaluate the performance of Browseragent, we selected six representative benchmark datasets, which are divided into two categories: The first category is general question answering datasets, including NQ (Kwiatkowski et al., 2019), and PopQA (Mallen et al., 2023); the second category is multi-hop question answering datasets, including HotpotQA (Yang et al., 2018), 2WikiMultiHopQA (Ho et al., 2020), Musique (Trivedi et al., 2022), and Bamboogle (Press

---

**Algorithm 1** Rollout/Inference Pipeline

---

**Require:** Input question $q$, BrowserAgent model $\pi_\theta$, VerlTool tool server $\mathcal{V}$, webpage $w$, observation from webpage $o$, history actions $A$, memory $M$, maximum action steps $S$, trace id $u_{id}$.

**Ensure:** Final answer $ans$.

1: Initialize webpage $w \leftarrow Wikipedia$
2: Initialize observation $o \leftarrow \mathcal{V}(w, a = \emptyset)$
3: Initialize history actions $A \leftarrow \emptyset$
4: Initialize memory $M \leftarrow \emptyset$
5: Initialize action count $s \leftarrow 0$
6: Initialize trace id $u_{id} \leftarrow rand()$
7: **while** $s < S$ **do**
8:     Generate response $y \sim \pi_\theta(q, o_s, A, M)$
9:     **if** `<conclusion> </conclusion>` detected in $y$ **then**
10:         Extract $m_s = \text{Parse}(y, $ `<conclusion>`,`</conclusion>`$)$
11:         Insert $m_s$ into memory $M \leftarrow M + m_s$
12:     **end if**
13:     **if** "`comand [parameters]`" detected in $y$ **then**
14:         Extract $a_s = \text{Parse}(y, $ "`","`"$)$
15:         Insert $a_s$ into history actions $A \leftarrow A + a_s$
16:         **if** `stop` equal to `comand` **then**
17:             **return** Final answer $ans = $ `parameters`
18:         **end if**
19:     **else**
20:         Set $a_s = \emptyset$
21:     **end if**
22:     Reset observation from webpage $o_{s+1} \leftarrow \mathcal{V}(w_s, a_s, u_{id})$
23:     Increment action count $s \leftarrow s + 1$
24: **end while**

---

et al., 2023). These datasets encompass a diverse range of search and reasoning tasks, enabling a systematic and comprehensive evaluation of Browseragent.

### 3.2 EVALUATION METRICS

In our experiments, we adopted two evaluation methods: **exact match (EM)** (Yu et al., 2024) and **LLM-based judgment**. EM offers an objective measure of consistency between model outputs and ground truth, but often underestimates performance when answers are factually correct and semantically reasonable yet differ in surface form (see case study in Appendix B.4). To complement this, we introduced an LLM-based judgment mechanism: each model-generated answer was independently evaluated by GPT-4.1 (OpenAI, 2024), Gemini Flash 2.5 (DeepMind, 2023), and Claude Sonnet 3.7 (Anthropic, 2024), and classified as "valid" if at least two judged it correct. By combining EM with consensus voting among multiple LLMs, our framework provides a more comprehensive and reliable view of model performance, mitigating the limitations of single-metric evaluation in complex, open-domain QA tasks. The full evaluation prompt is provided in Appendix B.2.

### 3.3 BASELINES

To evaluate the effectiveness of BrowserAgent, we compared it with the following baseline methods, adopting the same settings as those used in Search-R1 (Jin et al., 2025b): (1) Inference without Retrieval: Direct inference and Chain-of-Thought (CoT) reasoning (Wei et al., 2023). (2) Inference with Retrieval: Retrieval-Augmented Generation (RAG) (Lewis et al., 2021), IRCoT (Trivedi et al., 2023), and Search-o1 (Li et al., 2025d). (3) Fine Tuning-Based Methods: SFT (Chung et al., 2022); (4) RL-based methods: Search-R1 (Jin et al., 2025b).

These baselines cover a broad range of paradigms in retrieval augmentation and fine-tuning, providing a representative comparison for evaluating BrowserAgent. For fairness, we use the same pre-trained LLM across all methods and align the training data sources with those of the base-

lines. Although our training scale is much smaller than those used in prior work, BrowserAgent still achieves non-trivial performance gains.

## 3.4 EXPERIMENTAL SETUP

We conduct our experiments using the Qwen2.5-7b-Instruct model (Team, 2024). The 2022 Wikipedia dump (Karpukhin et al., 2020) is used as the knowledge source. During evaluation, we limit the model to a maximum of 30 rounds of browser interactions. Other training details can be found in the Appendix B.6.

## 3.5 EXPERIMENTAL RESULTS

Based on the experimental results in Table 2, we find that BrowserAgent consistently outperforms strong baselines across a range of tasks, demonstrating robust performance in both in-distribution and out-of-distribution evaluations. Our RFT stage achieves consistent improvements over the SFT stage, particularly on multi-hop QA tasks. Notably, BrowserAgent significantly outperforms multiple versions of Search-R1. Specifically, BrowserAgent-7B can achieve around a **20%** improvement over Search-R1-Instruct on multi-hop QA tasks. This advantage stems from its ability to handle longer reasoning chains without being constrained by context length. Even with limited training data and only a simple, efficient RFT strategy, it still outperforms Search-R1-Instruct trained with complex reinforcement learning techniques, fully demonstrating the effectiveness and practicality of BrowserAgent's end-to-end paradigm based on direct interaction with web pages in tasks involving active web exploration and complex reasoning. Moreover, as RFT does not use explicit format-based rewards, EM alone may not fully capture performance; incorporating LLM-based judgment provides a more comprehensive evaluation of the model's true capabilities.

Table 2: Main results of baselines and our methods. The best-performing results for EM and LLM-based judgment are highlighted in bold.

| Methods | General QA | | Multi-Hop QA | | | | Avg. |
|---|---|---|---|---|---|---|---|
| | NQ | PopQA | HotpotQA | 2wiki | Musique | Bamboogle | |
| **IND/OOD** | IND | OOD | IND | OOD | OOD | OOD | |
| Search-R1-Direct Inference | 0.134 | 0.140 | 0.183 | 0.250 | 0.031 | 0.120 | 0.143 |
| Search-o1 | 0.151 | 0.131 | 0.187 | 0.176 | 0.058 | 0.296 | 0.167 |
| Search-R1-CoT | 0.048 | 0.054 | 0.092 | 0.111 | 0.022 | 0.232 | 0.093 |
| Search-R1-IRCoT | 0.224 | 0.301 | 0.133 | 0.149 | 0.072 | 0.224 | 0.184 |
| Search-R1-RAG | 0.349 | 0.392 | 0.299 | 0.235 | 0.058 | 0.208 | 0.257 |
| Search-R1-SFT | 0.318 | 0.121 | 0.217 | 0.259 | 0.066 | 0.112 | 0.182 |
| Search-R1-Instruct | **0.393** | 0.397 | 0.370 | 0.414 | 0.146 | 0.368 | 0.348 |
| **Ours** | | | | | | | |
| Qwen2.5-7b-Instruct (EM) | 0.146 | 0.244 | 0.138 | 0.097 | 0.039 | 0.080 | 0.124 |
| Browseragent-SFT (EM) | 0.371 | **0.437** | 0.441 | **0.500** | 0.157 | 0.456 | 0.394 |
| Browseragent-RFT (EM) | 0.388 | 0.431 | **0.458** | 0.498 | **0.164** | **0.504** | **0.407** |
| Qwen2.5-7b-Instruct (LLM-judge) | 0.191 | 0.279 | 0.186 | 0.154 | 0.054 | 0.096 | 0.160 |
| Browseragent-SFT (LLM-judge) | 0.466 | **0.487** | 0.547 | 0.599 | 0.204 | 0.504 | 0.468 |
| Browseragent-RFT (LLM-judge) | **0.493** | 0.485 | **0.561** | **0.601** | **0.212** | **0.552** | **0.484** |

## 3.6 ABLATION

For the ablation studies, we randomly sampled 1K examples from each benchmark dataset for evaluation. As Bamboogle contains fewer than 1,000 instances, we evaluated it in full. Since RFT models trained on poorly performing SFT models tend to exhibit even lower performance, we conducted most ablation experiments on the SFT model, while using the RFT model specifically for step-wise ablations. In this section, we report only the EM scores as the evaluation metric.

**Increasing Memory Capacity.** For instance, when answering a question like "Who is the father of the greatest NBA player of all time?", the model may first determine that "Michael Jordan is the

greatest NBA player" and then retrieve that "Jordan's father is James Jordan." Without storing the intermediate conclusion, the model risks losing the connection to the original question and falling into a circular search pattern. To enhance reasoning efficiency and avoid redundant steps, we increased the model's memory capacity by training it to summarize and retain intermediate conclusions during multi-hop reasoning. The experimental results in Table 3 show that the proposed method not only effectively prevents information loss and logical disconnection in reasoning chains, but also improves the model's overall performance and generalization when handling complex tasks.

Table 3: Results of the Ablation Study on BrowserAgent-SFT.

| Memory | Single-round | Steps | Parameters | General QA | | Multi-Hop QA | | | | Avg. |
|--------|--------------|-------|------------|------------|------|--------------|------|---------|-----------|------|
| | | | | NQ | PopQA | HotpotQA | 2wiki | Musique | Bamboogle | |
| | | | | IND | OOD | IND | OOD | OOD | OOD | |
| ✓ | ✗ | 6 | 7B | 0.316 | 0.420 | 0.382 | 0.385 | 0.130 | 0.368 | 0.334 |
| ✓ | ✓ | 6 | 3B | 0.298 | 0.362 | 0.335 | 0.359 | 0.087 | 0.264 | 0.284 |
| ✓ | ✓ | 6 | 7B | 0.332 | 0.420 | 0.384 | 0.423 | 0.131 | 0.360 | 0.342 |
| ✗ | ✓ | 30 | 7B | 0.341 | 0.417 | 0.399 | 0.415 | 0.132 | 0.384 | 0.348 |
| ✓ | ✓ | 30 | 7B | 0.387 | 0.466 | 0.434 | 0.457 | 0.152 | 0.456 | 0.392 |

**Multi-round observation v.s. Single-round observation** Previous argentic posttraining works such as RAGEN (Wang et al., 2025) and Search-R1 treat the entire tool-use trajectory as a single input, appending all retrieved content and intermediate reasoning steps into the context. This approach is limited by the context window, restricting scalability during inference. To address this, we introduce a memory mechanism that stores only key conclusions in a structured <memory> format, removing redundant context while preserving essential information. This enables the model to reason more efficiently and scale to up to 30 reasoning steps without exceeding context limits. The experimental results in Table 3 demonstrate that trimming redundant input not only improves model performance, but also enhances its scalability in handling complex multi-hop tasks.

**Model Size Scaling** Due to computational resource constraints, we conducted comparative experiments using models with 3B and 7B parameters. As shown in the Table 3, the 7B model consistently outperforms the 3B model across all evaluation benchmarks, demonstrating stronger reasoning capabilities and better generalization. This result provides compelling evidence for the importance of scaling model size in multi-hop reasoning and complex question answering tasks. Furthermore, it highlights that even under limited training data and relatively lightweight fine-tuning procedures, model capacity remains a critical factor influencing overall performance.

**Test Step Scaling** Leveraging our single-turn context window architecture, the proposed method scales effectively by increasing the number of interaction rounds. While performance is slightly lower than Search-R1 at around 6 turns, it improves steadily as the number of turns grows. Notably, even with a significantly increased maximum step limit (e.g., 30 steps), the average number of reasoning steps remains stable, indicating the model's ability to complete tasks efficiently without overusing the step budget. The detailed experimental results can be found in Tables 4 and 5 in the Appendix B.7.1.

## 4 RELATED WORK

### 4.1 REJECTION SAMPLING FINE-TUNING

Rejection Sampling Fine-Tuning (Dong et al., 2023) is a widely adopted strategy for enhancing large language models (LLMs), particularly when integrating generative models with external evaluation criteria or retrieval mechanisms. In this approach, the model generates multiple candidate outputs (Yuan et al., 2023; Xiong et al., 2025; Huang et al., 2022) for each input. These candidates are then assessed using predefined evaluation metrics—such as answer correctness or adherence to specific constraints (Yang et al., 2024). Only the top-performing responses, as determined by the evaluation criteria, are retained through rejection sampling. The resulting high-quality samples are subsequently used to further fine-tune the model (Zelikman et al., 2022). This method has been shown to effectively improve the accuracy and robustness of LLMs across various tasks, including

question answering, code generation (Li et al., 2025a), and reasoning (Team, 2025). We extend this method to the domain of browser agents and achieve strong performance on several key tasks, demonstrating its broad applicability and effectiveness in dynamic web environments.

## 4.2 AGENTIC INFORMATION SEEKING

Agentic Information Seeking is a retrieval approach in which an agent actively participates in acquiring and integrating information. Unlike traditional keyword search, this method allows the agent to behave like a human user, automatically performing a series of operations in a browser or web environment—such as typing, clicking, and navigating—to progressively collect and filter useful information. (de Chezelles et al., 2025; Nakano et al., 2022; Schick et al., 2023; Jiang et al., 2025; Murphy et al., 2025) It is typically powered by large language models or automation strategies, enabling the agent to understand complex tasks, dynamically adjust its actions, and synthesize information from multiple sources to meet complex or multi-turn information needs. Although current approaches (Jin et al., 2025b; Wu et al., 2025) make notable progress in this field, they still face several limitations—for instance, their limited capacity to model long-context information, which results in a heavy reliance on external resources such as the Jina service and larger-scale pretrained models. To address these challenges, we propose a strategy that integrates web page scrolling operations (e.g., scroll up and down) into an end-to-end model, enabling the agent to actively read web content. This approach effectively alleviates the aforementioned issues and enables efficient understanding of complex web information.

## 4.3 AGENTIC SEARCH

Web Agents are intelligent agent systems capable of autonomously interacting with internet resources (Li et al., 2025e;b; Liu et al., 2025b), such as web pages and search engines, to accomplish complex tasks including multi-step reasoning, information retrieval, question answering, and data integration. Typically, these systems leverage LLMs as their decision-making core, in combination with tool invocation, web page parsing, retrieval, and automated execution functionalities. In recent years, Web Agents have become a critical foundation for research in open-domain problem solving, retrieval-augmented generation (Chen et al., 2025b), AI assistants, and automated search-based reasoning. Recent works mainly focus on benchmarking (Zhou et al., 2024; Yao et al., 2023a; Koh et al., 2024; Yoran et al., 2024; Yang et al., 2025) rather than training agents directly in these environments, primarily because browser-based interaction is computationally expensive. To this end, we have implemented an efficient and easy-to-use two-stage browser agent training pipeline, which ensures scalability while significantly reducing the system overhead of environment interactions, thereby laying the foundation for large-scale autonomous learning of Web Agents.

## 5 CONCLUSION AND FUTURE WORK

This paper presents BrowserAgent, an efficient and scalable training framework for real-world web interaction. Unlike previous approaches that rely on static summaries or external parsing services, our method leverages Playwright to enable end-to-end web operations, allowing the model to actively acquire and integrate information through fine-grained actions such as scrolling and clicking. Combined with a parallel tool serving architecture and an explicit memory mechanism, our two-stage SFT + RFT training strategy reduces redundant context while enabling efficient scaling along complex reasoning chains. Experimental and ablation results demonstrate that BrowserAgent achieves significant advantages in performance, scalability, and data efficiency. Notably, BrowserAgent outperforms existing methods (e.g., Search-R1) using a smaller amount of training data, especially excelling in multi-hop reasoning tasks.

In future work, we plan to explore more intelligent memory mechanisms, cross-website generalization, multi-agent collaboration, and continual learning from interaction logs to advance BrowserAgent toward a truly general-purpose web agent.

ETHICS STATEMENT

This study does not involve any personal data, sensitive information, or high-risk application scenarios. No ethically controversial datasets or models were used. All experimental data are standard benchmark datasets that are publicly available, and the sole purpose of this research is to advance the development of web-based agent systems. Therefore, we believe this work does not pose any significant ethical risks.

REPRODUCIBILITY STATEMENT

To ensure the reproducibility of our experiments, we have provided the complete implementation code in the supplementary materials. All technical details, including the evaluation benchmarks, baseline methods, and training hyperparameter settings used in this work, can be found in Section 3.

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

## A  THE USAGE OF LLMS

In this study, we use LLMs for data generation, result evaluation in the experiments. For the paper writing, we use LLMs for paper polishing, generating visualizations like figures, and retrieving related work. We have proofread carefully to ensure no hallucinated content in the paper.

## B  APPENDIX

### B.1  EVAL LOSS CURVE

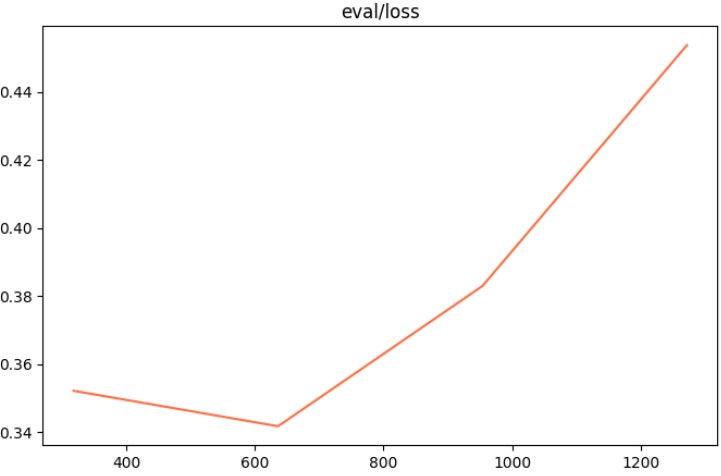

Figure 4: The evaluation loss curve of SFT.

### B.2  EVALUATION PROMPT

> **Eval Prompt**
>
> You are an agent tasked with determining the correctness of an answer. Given a question, its corresponding ground truth, and an answer, you need to decide whether the answer is correct. If it is correct, please output "yes" otherwise output "no".
> question: {}
> ground truth: {}
> answer: {}

## B.3 SYSTEM PROMPT

---

**Prompt**

You are a browser interaction assistant designed to execute step-by-step browser operations efficiently and precisely to complete the user's task. You are provided with specific tasks and webpage-related information, and you need to output accurate actions to accomplish the user's task.

Here's the information you'll have:

The user's objective: This is the task you're trying to complete.

The current web page's accessibility tree: This is a simplified representation of the webpage, providing key information.

The open tabs: These are the tabs you have open.

The previous actions: There are the actions you just performed. It may be helpful to track your progress.

Information already found: Information related to the current query that has been identified in historical actions. You need to integrate and supplement this information.

The actions you can perform fall into several categories:

Page Operation Actions:

'click [id] [content]': This action clicks on an element with a specific id on the webpage.

'type [id] [content] [press_enter_after=0|1]': Use this to type the content into the field with id. By default, the ""Enter"" key is pressed after typing unless press_enter_after is set to 0.

'hover [id] [content]': Hover over an element with id.

'press [key_comb]': Simulates the pressing of a key combination on the keyboard (e.g., Ctrl+v).

'scroll [down|up]': Scroll the page up or down.

Tab Management Actions:

'new_tab': Open a new, empty browser tab.

'tab_focus [tab_index]': Switch the browser's focus to a specific tab using its index.

'close_tab': Close the currently active tab.

URL Navigation Actions:

'goto [url]': Navigate to a specific URL.

'go_back': Navigate to the previously viewed page.

'go_forward': Navigate to the next page (if a previous 'go_back' action was performed).

Completion Action:

'stop [answer]': Issue this action when you believe the task is complete. If the objective is to find a text-based answer, provide the answer in the bracket. If you believe the task is impossible to complete, provide the answer as ""N/A"" in the bracket.

To be successful, it is very important to follow the following rules:

1. You should only issue an action that is valid given the current observation.

2. You should only issue one action at a time.

3. You should follow the examples to reason step by step and then issue the next action.

4. You should refer to historical actions when issue an action and try not to make repetitive actions.

5. All reasoning must be inside '<think></think>' tags, and there must be no output before '<think></think>'.

6. After '<think></think>', only the action should be generated in the correct format, enclosed in code fences. For example:

<think>This button looks relevant to my goal. Clicking it should take me to the next step.</think>

'''click [id] [content]'''

7. Issue the stop action when you think you have achieved the objective. Don't generate anything after stop.

8. Always format actions correctly:

'''command [parameters]'''

For example, if searching for ""death row inmates in the US"" in a search field with ID '21', correctly format it as:

'''type [21] [death row inmates in the US] [1]'''

Avoid incorrect formats that omit brackets around parameters or numeric values.

9.Between <think></think>, you need to use <conclusion></conclusion> to enclose the information obtained in this round that is relevant to the current query. Note that if there is no valid information, this part is not required. The enclosed information must be directly usable to answer the original query.

---

## B.4 CASE STUDY

In this case of the NQ dataset, our BrowserAgent uses 3 steps to interact with the web environment and find the final answer of the given question. The ground truth of the question is "Edward IV of England", but the BrowserAgent gives the final answer "King Edward IV". Obviously, these two are semantically equivalent, but we cannot use the EM method to match them.

**Case Study**

**Question:** Who was the father of the princes in the tower?

| Step 1 |
|---|

**Input Sequence:**

System prompt: ...

Observation:

```
    ...
    [331] textbox "Search 'Wikipedia'" required: False
    [333] checkbox '' checked: false
    [340] link 'Go to welcome page' url: https://tigerai.ca/
    [342] link "Go to the main page of 'Wikipedia'" url:
    ↪  https://tigerai.ca/wikipedia_en_all_maxi_2022-05/
    [343] button 'Wikipedia'
    [344] link 'Go to a randomly selected page' url:
    ↪  https://tigerai.ca/random?content=wikipedia_en_all_maxi_2022-05
    ...
```

History_action:

History_info:

**Output Sequence:**

*<think>*
The user's objective is to find out who was the father of the Princes in the Tower. The current page is a landing page with a search box labeled "Search 'Wikipedia'" (id: 331). The most efficient way to find the answer is to use the search function to look up "Princes in the Tower", as this will likely lead to an article that mentions their father. I will type the query into the search box and press enter.
*</think>*

Action:

```
    type [331] [Princes in the Tower] [1]
```

| Step 2 |
|---|

**Input Sequence:**

System prompt: ...

Observation:

```
    ...
    [742] StaticText 'Results 1-25 of 40,000 for "Princes in the
    ↪  Tower"'
    [1459] link 'Princes in the Tower' url:
    ↪  https://tigerai.ca/wikipedia_en_all_maxi_2022-05/A/...
    [790] StaticText 'from Wikipedia'
    [791] StaticText '8,257 words'
    [1490] link 'Shin Yokohama Prince Hotel' url:
    ↪  https://tigerai.ca/wikipedia_en_all_maxi_2022-05/A/...
    [793] StaticText '...The Shin Yokohama Prince Hotel is a skyscraper
    ↪  in the Kohoku-ku ward of Yokohama, Japan. Completed in March
    ↪  1992, it stands at 149.4m (490 ft) tall.
    ...
```

History_action:

```
    type [331] [Princes in the Tower] [1]
```

History_info:

**Output Sequence:**

*<think>*
The user's objective is to find out who was the father of the princes in the tower. The search results page shows a link titled "Princes in the Tower," which is likely to contain relevant information about the princes

and their father. Clicking this link should lead to the main article where the answer is probably found.
*</think>*

*<conclusion>*
There is a link to the Ṗrinces in the Towerärticle, which is likely to contain information about their father.
*</conclusion>*

Action:

```
click [1459] [Princes in the Tower]
```

**Step 3**

**Input Sequence:**

System prompt: ...

Observation:

```
    ...
    [3655] link 'Richard of Shrewsbury, Duke of York' url:
    ↪  https://tigerai.ca/wiki/wikipedia_en_all_maxi_2022-05/A/...
    [2285] StaticText 'These two brothers were the only sons of King'
    [3656] link 'Edward IV' url:
    ↪  https://tigerai.ca/wiki/wikipedia_en_all_maxi_2022-05/A/...
    [2287] StaticText ' and '
    [3657] link 'Elizabeth Woodville' url:
    ↪  https://tigerai.ca/wiki/wikipedia_en_all_maxi_2022-05/A/...
    ...
```

History_action:

```
    type [331] [Princes in the Tower] [1]
    click [1459] [Princes in the Tower]
```

History_info:
    There is a link to the "Princes in the Tower" article, which is likely to contain information about their father.

**Output Sequence:**

*<think>*
The static text on the page states: "These two brothers were the only sons of King Edward IV and Elizabeth Woodville." This directly answers the question about the father of the Princes in the Tower.
*</think>*

*<conclusion>*
The father of the Princes in the Tower was King Edward IV.
*</conclusion>*

Action:

```
    stop [King Edward IV]
```

**Final answer:** King Edward IV
**Ground truth:** Edward IV of England

In this case of the HotpotQA dataset, our BrowserAgent uses 3 steps to interact with the web environment and find the final answer of the given question. The ground truth of the question is "no", but the BrowserAgent gives the final answer "Skin Yard was from the U.S. (Seattle, Washington). Ostava was from Bulgaria (not the U.S.).". For this judgmental question, our agent provided a negative answer to prove that the two cities come from different countries, which is equivalent to a negative answer from the ground truth.

**Case Study**

**Question:** Were the bands Skin Yard and Ostava from the U.S.?

**Step 1**

**Input Sequence:**

System prompt: ...

Observation:

```
    ...
    [333] textbox \"Search 'Wikipedia'\" required: False
    [335] checkbox '' checked: false
    [342] link 'Go to welcome page' url: https://tigerai.ca/
    [344] link \"Go to the main page of 'Wikipedia'\" url:
    ↪  https://tigerai.ca/wikipedia_en_all_maxi_2022-05/
    [345] button 'Wikipedia'
    [346] link 'Go to a randomly selected page' url:
    ↪  https://tigerai.ca/random?content=wikipedia_en_all_maxi_2022-05
    ...
```

History_action:

History_info:

**Output Sequence:**

*<think>*
To determine whether Skin Yard and Ostava were from the U.S., I need to look up each band individually. The most efficient way is to use the Wikipedia search box available on this page. I will start by searching for "Skin Yard" using the search textbox with id [333].
*</think>*

Action:

```
    type [333] [Skin Yard] [1]
```

**Step 2**

**Input Sequence:**

System prompt: ...

Observation:

```
    ...
    [743] StaticText 'Results 1-25 of 4,000 for \"Skin Yard\"'
    [1155] link 'Skin Yard' url:
    ↪  https://tigerai.ca/wikipedia_en_all_maxi_2022-05/A/Skin_Yard
    [750] StaticText 'Skin Yard Skin Yard was an American grunge band
    ↪  from Seattle, Washington, who were active from 1985 to 1992.
    ↪  The group never gained a mainstream audience, but were an
    ↪  influence on several of their grunge contemporaries, including
    ↪  Soundgarden, Screaming Trees, and Green River. Skin Yard
    ↪  Publicity photo 1988, feat. Jack Endino, Ben McMillan, and
    ↪  Daniel House Background information Origin Seattle, Washington,
    ↪  U.S. Genres Grunge[1] alternative metal Years active 1985-1992
    ↪  Labels Cruz, Toxic Shock......'
    ...
```

History_action:

```
    type [333] [Skin Yard] [1]
```

History_info:

**Output Sequence:**

*<think>*
We have clear evidence from the static text that Skin Yard was an American grunge band from Seattle, Washington, U.S. This answers the first part of the question. Now, we need to check if Ostava was from the U.S. The current page is focused on Skin Yard, so I should search for Ostava next. The search box has id [1133], so I will type "Ostava" into it and search.
*</think>*

*<conclusion>*
Skin Yard was from Seattle, Washington, U.S.
*</conclusion>*

Action:

```
    type [1133] [Ostava] [1]
```

**Step 3**

**Input Sequence:**

System prompt: ...

Observation:

```
    ...
    [1455] StaticText 'Results 1-24 of 24 for \"Ostava\"'
    [1630] link 'Ostava' url:
    ↪  https://tigerai.ca/wikipedia_en_all_maxi_2022-05/A/Ostava
    [1462] StaticText 'Ostava Ostava is an alternative rock band from
    ↪  Bulgaria. It was formed in 1991, and released a long-playing
    ↪  record in 2000. Ostava Background information Origin Gabrovo,
    ↪  Bulgaria Genres Alternative rock Years active 1991-present
    ↪  Labels Riva Sound Stars records Stain Studio Virginia
    ↪  Records/Universal Members Svilen Noev Georgi Georgiev Daniel
    ↪  Petrov Boyan Petkov Mihail Shishkov Alexander Marburg History
    ↪  Formation and early years Fender Jazz Bass played by Boyan
    ↪  Petkov Ostava was formed in 1991......'
    ...
```

History_action:

```
    type [333] [Skin Yard] [1]
    type [1133] [Ostava] [1]
```

History_info:
    Skin Yard was from Seattle, Washington, U.S.

**Output Sequence:**

*<think>*
We already know from HISTORY_info that Skin Yard was from Seattle, Washington, U.S. Now, from the current observation, we see that Ostava is described as "an alternative rock band from Bulgaria. It was formed in 1991..." and its origin is listed as Gabrovo, Bulgaria. Therefore, Ostava is not from the U.S., but from Bulgaria.
 *</think>*

*<conclusion>*
Skin Yard was from the U.S. (Seattle, Washington). Ostava was from Bulgaria (not the U.S.).
 *</conclusion>*

Action:

```
    stop [Skin Yard was from the U.S. (Seattle, Washington). Ostava was
    ↪  from Bulgaria (not the U.S.).]
```

**Final answer:** Skin Yard was from the U.S. (Seattle, Washington). Ostava was from Bulgaria (not the U.S.).
**Ground truth:** no

## B.5 FILTERING STRATEGY AND DATA RATIO

We used string matching to filter out RFT data that did not conform to our expected format. To preserve the original data distribution and mitigate the catastrophic forgetting of answer formatting learned during the SFT stage, we randomly sampled 80% of the SFT data to include in the final dataset used for the second-stage training. To ensure consistency in data volume and proportion between the two stages, we randomly selected 400 NQ samples and 673 HotpotQA samples from the filtered Original RFT data, resulting in a total of 1,073 representative reasoning examples. By combining these 1,073 samples with the 80% subset of SFT data, we obtained the final dataset used for RFT training.

## B.6 EXPERIMENTAL SETUP

During the training phase, we adopt the 5.3K synthesized data for SFT and RFT. Both training and evaluation were conducted on 8 NVIDIA A100 GPUs. The SFT stage was performed for 2 epochs with a learning rate of 5e-6, while the RFT stage was also trained for 2 epochs, using a lower learning rate of 1e-6. Model evaluation was conducted on the test or validation sets of six publicly available question answering datasets, covering a diverse range of tasks to systematically assess the model's generalization ability in both in-domain and out-of-domain settings. The evaluation metrics include EM and LLM-based judgment, providing a comprehensive reflection of the model's accuracy and real-world applicability.

## B.7 EXPERIMENTAL RESULTS

### B.7.1 TEST STEP SCALING

Table 4: Results of the Ablation Study on BrowserAgent-RFT.

| Steps | General QA | | Multi-Hop QA | | | | Avg. |
|---|---|---|---|---|---|---|---|
| | NQ | PopQA | HotpotQA | 2wiki | Musique | Bamboogle | |
| | IND | OOD | IND | OOD | OOD | OOD | |
| 6 | 0.361 | 0.399 | 0.402 | 0.409 | 0.133 | 0.352 | 0.343 |
| 15 | 0.384 | 0.404 | 0.419 | 0.449 | 0.145 | 0.480 | 0.380 |
| 30 | 0.390 | 0.465 | 0.465 | 0.457 | 0.161 | 0.504 | 0.407 |

Table 5: Average number of steps in the ablation study on BrowserAgent-RFT.

| Steps | General QA | | Multi-Hop QA | | | | Avg. |
|---|---|---|---|---|---|---|---|
| | NQ | PopQA | HotpotQA | 2wiki | Musique | Bamboogle | |
| | IND | OOD | IND | OOD | OOD | OOD | |
| 6 | 3.039 | 2.799 | 3.442 | 4.022 | 4.128 | 3.909 | 3.557 |
| 15 | 3.516 | 3.248 | 3.979 | 4.999 | 4.648 | 4.400 | 4.132 |
| 30 | 3.933 | 3.856 | 3.856 | 4.945 | 4.638 | 4.492 | 4.287 |

### B.7.2 TRIVIAQA RESULTS

As shown in the Table 6, there is a significant discrepancy between the EM scores and LLM-judge evaluation scores for our two models on the TriviaQA dataset. We attribute this gap to the absence of format-specific supervision signals in our training process, which leads to answers that are factually correct but fail to satisfy exact match criteria due to formatting issues. For detailed examples, please refer to the case study in the Appendix B.4.

Table 6: TriviaQA Results.

| Model | TriviaQA |
|---|---|
| Browseragent-SFT (EM) | 0.206 |
| Browseragent-RFT (EM) | 0.206 |
| Browseragent-SFT (LLM-judge) | 0.598 |
| Browseragent-RFT (LLM-judge) | 0.600 |

