# OpenReview forum: "BrowserAgent: Building Web Agents with Human-Inspired Web Browsing Actions"
_ICLR.cc/2026/Conference — ICLR 2026 Conference Withdrawn Submission_

### Official Review · Reviewer_K2qk · 2025-10-16

**Soundness:** 2
**Presentation:** 2
**Contribution:** 2
**Rating:** 2
**Confidence:** 5

**Summary:**

They propose BrowserAgent, a novel framework for web agents that more closely mimics human browsing behavior by interacting directly with dynamic web environments. Unlike prior state-of-the-art models such as Search-R1, which depend on external tools to convert web pages into static text, BrowserAgent operates on raw web content through a predefined set of browser actions (e.g., scroll, click, type) implemented via the Playwright library.

This paradigm first employs Supervised Fine-Tuning (SFT) to teach the agent basic interaction patterns, followed by Rejection Fine-Tuning (RFT) to help it learn from suboptimal action sequences. This approach proves to be remarkably data-efficient, enabling BrowserAgent to achieve superior generalization and performance despite using significantly less training data than its predecessors.

They addresses the challenge of long-horizon tasks by incorporating an explicit memory mechanism. This module allows the agent to store key conclusions and information gathered across multiple interaction steps, substantially enhancing its multi-hop reasoning capabilities.

**Strengths:**

The most significant strength is the move away from reliance on external tools (like Jina services or GPT-4o for summarization). The agent operates directly on raw web pages via Playwright, using a set of human-inspired actions (click, scroll, type, tab management). This allows it to access more fine-grained, dynamic information and avoids the information loss that occurs when converting a web page into static text.

**Weaknesses:**

* The experiments are conducted on a locally hosted, offline snapshot of Wikipedia from 2022 (using Kiwix). This is a controlled, clean, and ad-free environment with a relatively consistent structure. It is not representative of the real, "messy" internet, which is filled with dynamic ads, pop-ups, CAPTCHAs, complex JavaScript interactions, and diverse layouts.

* The training data (5.3K trajectories) is generated by having GPT-4.1 interact with the environment. This means BrowserAgent is essentially learning to "imitate" the browsing strategy of GPT-4.1. This approach caps the agent's potential, as it is unlikely to discover novel strategies that surpass its teacher's capabilities. It may also inherit any biases or suboptimal behaviors of the teacher model.

**Questions:**

1. The paper fails to demonstrate the performance advantages of a browser-based agent (like BrowserAgent) compared to API-based web agents. This comparison is completely missing from the manuscript.

2. The baselines used for comparison are outdated. The authors should evaluate their model on more recent and comprehensive search benchmarks, such as BrowseComp, X-BENCH, or GAIA.

3. The set of baselines is not only limited but also relatively weak. To better situate the paper's contributions, it is essential to compare against stronger and more relevant state-of-the-art methods, such as WebDancer, Deepdive, or WebSailor.

4. The use of a memory module for context management during inference raises a concern about latency. Since the context is modified at each step, this could lead to a decreased KV cache hit rate, resulting in higher inference latency. Have the authors analyzed this potential performance trade-off?

5. The methods for data synthesis and training (SFT + RFT) lack significant novelty and follow standard practices. Have the authors considered or experimented with Reinforcement Learning (RL) to further optimize the agent's policy, especially given that some competing baselines utilize it?

---

### Official Review · Reviewer_KMVj · 2025-10-26

**Soundness:** 2
**Presentation:** 2
**Contribution:** 2
**Rating:** 4
**Confidence:** 4

**Summary:**

BrowserAgent proposes a browser-native agent that operates directly on live web pages using human-like atomic actions (click, scroll, type, etc.), combined with a lightweight memory mechanism and a two-stage post-training process (SFT and Rejection Fine-Tuning). The paper claims improved reasoning efficiency and lower operational cost compared to prior web agents that depend on multi-tool pipelines or reinforcement learning. While the approach is innovative in its design simplicity and interpretable memory use, it lacks solid empirical validation, particularly regarding comparisons to true WebAgent systems, cost efficiency evidence, and the quantitative impact of its claimed advantages.

**Strengths:**

1. The framework’s browser-native design avoids intermediate summarizers and external APIs, resulting in a more human-like browsing behavior. This simplifies the web interaction pipeline and makes the research results closer to real-world browsing applications.
2. By relying on SFT + RFT instead of complex RL methods like PPO or DPO, the training pipeline is reproducible and less resource-intensive. The results show its advantages on multiple QA benchmarks even without RL-training.

**Weaknesses:**

1. Although the authors claim that WebAgent-style methods are expensive and less flexible, no actual webagent baseline is evaluated. This makes the cost-efficiency and interaction limitations claims speculative rather than evidence-based.

2. The paper asserts that avoiding tool calls lowers cost but provides no quantitative analysis, without runtime, API call frequency, token usage, inference cost measurements.

3. Search-R1 seems to be evaluated under a different environment than its original setup, leading to lower performance. The lack of detailed implementation or adaptation notes limits interpretability of the comparative results and weakens fairness claims.

4. The “memory” is effectively a textual note-taking process similar to the ReAct reflection/summarization pattern, without architectural innovation.

**Questions:**

1. Can the authors provide quantitative comparisons (token usage, API calls, inference time) to substantiate the claim that BrowserAgent is lower-cost than WebAgent systems?

2. How are Search-R1 baselines implemented in this paper? The original results in different QA tasks are much higher than those reported in this paper. Are there implementation constraints that may disadvantage them relative to BrowserAgent?

3. Why were no direct baselines of webagent included, given that the paper critiques them in the introduction?

4. Will the memory mechanism bring extra inference cost? What is the difference of the proposed memory mechanism with context summarization and reflection.

5. Why TriviaQA comparison is missing (only partial results in appendix) while it is a important bnehcmark for general QA in Search-R1?

---

### Official Review · Reviewer_CZep · 2025-11-05

**Soundness:** 3
**Presentation:** 3
**Contribution:** 3
**Rating:** 6
**Confidence:** 2

**Summary:**

The paper introduces BrowserAgent, an interactive web agent designed to solve complex, real-world problems by browsing the web. Unlike previous agents that rely on external tools to convert web pages into static text, BrowserAgent operates directly on raw web pages. It uses a set of human-inspired actions—such as scrolling, clicking, and typing—to navigate and interact with dynamic web environments via the Playwright framework.

To train the agent, the authors use a lightweight two-stage training pipeline:
1. Supervised Fine-Tuning (SFT): This initial stage teaches the base model (Qwen-7B-Instruct) basic reasoning capabilities and the correct format for its thoughts and actions.
2. Rejection Fine-Tuning (RFT): In this second stage, the SFT model generates multiple potential reasoning paths. The system then selects the best correct answer (specifically, the one with the most reasoning steps) to further fine-tune the model, encouraging deeper and more robust reasoning.

A key feature of BrowserAgent is its explicit memory mechanism. This allows the agent to store key conclusions it finds during its browsing, helping it to maintain context and reason effectively over long-horizon, multi-step tasks.

**Strengths:**

It proposes a framework where the agent interacts directly with raw web pages using fine-grained actions like click, scroll, and type, which is a departure from relying on static text summaries.

The paper introduces an efficient SFT and RFT training strategy that improves the model's reasoning abilities without complex reinforcement learning techniques.

It introduces a memory module that stores key findings, significantly enhancing the agent's performance on complex multi-hop reasoning tasks.

 BrowserAgent-7B achieves significant performance gains over strong baselines like Search-R1, notably showing around a 20% improvement on multi-hop QA tasks (such as HotpotQA, 2Wiki, and Bamboogle) while using significantly less training data.

**Weaknesses:**

1. The agent operates on an "accessibility tree" parsed by Playwright, which is "pure text-based". This is a known limitation for all agents of this type. It means the agent is blind to visual layout, non-text elements (like complex JavaScript-rendered charts), and website designs that do not have a clean, descriptive accessibility tree.

2. The agent uses a "minimal yet expressive" set of predefined actions (e.g., click, type, scroll). This fixed set may not be sufficient for more complex, dynamic web interactions like drag-and-drop, handling complex pop-up modals, or solving CAPTCHAs, which are not mentioned in the action list.

3. Scalability of Training Data: While the authors developed a system to parallelize Playwright instances and improve data collection throughput, the process is still inherently complex. The training was conducted on only 5.3K trajectories, which is noted as being "significantly less" than baselines, but also points to the difficulty of creating this type of high-quality, browser-native data at scale.

**Questions:**

Above

---

### Note · Authors · 2025-11-13

I have read and agree with the venue's withdrawal policy on behalf of myself and my co-authors.